# Relationship between initial self-perceived depressive symptoms and disease severity in working patients with first-onset major depressive disorder

**Tomoyuki Hirota, Yasuhiko Deguchi** [ORCID]*, **Shinichi Iwasaki** [ORCID]**, Aya Sakaguchi, Akihiro Niki, Yoshiki Shirahama, Yoko Nakamichi, Koki Inoue**

Department of Neuropsychiatry, Graduate School of Medicine, Osaka City University, Osaka, Japan

* m2012567@med.osaka-cu.ac.jp

**Data Availability Statement:** All relevant data are within the paper.

**Funding:** The authors received no specific funding for this work.

## Abstract

The severity of major depressive disorder (MDD), which is related to the depressive symptoms, is a predictor of clinical outcomes and may be used to determine the appropriate treatment. However, there is a lack of systematic research on the relationship between early depressive symptoms and MDD severity. This study aimed to clarify the association between initial depressive symptoms and MDD severity in working patients. We assessed 118 patients aged over 20 years who visited the Neuropsychiatry Department of the Osaka City University Hospital following their first episode of MDD. Logistic regression analyses were performed to estimate the odds ratios (OR) with 95% confidence intervals (CI) for the associations between age, gender, marital status, working hours, and initial self-perceived depressive symptoms and MDD severity. Age and working hours were analyzed as continuous variables, and gender (man, woman), marital status (married, single) and severity (mild to moderate MDD, severe to very severe MDD) were analyzed as categorical variables. The most common initial self-perceived symptom was "depressed mood," followed by "fatigue or loss of energy nearly every day." The univariate analysis found no association between age, gender, marital status, or working hours and MDD severity. Initial self-perceived non-somatic symptoms were associated with increased odds of having severe MDD (odds ratio = 3.32, 95% confidence interval 1.46–7.58), and this association persisted in the adjusted model (odds ratio = 3.35, 95% confidence interval 1.47–7.60). Initial self-perceived non-somatic depressive symptoms are significantly associated with MDD severity at its first onset. Workplace support may lead to the early detection and treatment of working patients with non-somatic symptoms.

## Introduction

Major depressive disorder (MDD) is the third leading cause of financial burden and the leading cause of disability among all illnesses, with its lifetime prevalence estimated at 10% globally

**Competing interests:** The authors have declared that no competing interests exist.

and 6.5% in Japan [1–4]. Surveys by the Japanese Ministry of Health, Labor and Welfare have shown that the number of patients with mood disorders, including MDD, had increased from approximately 0.9 million to approximately 1.2 million in the last 15 years, and that the number of working-aged patients (under the age of 65 years) continues to increase [5]. MDD has been found to impact productivity and represents one of the major causes of workplace absenteeism and presenteeism [6]. Absenteeism associated with MDD was more prolonged and resulted in a higher annual cost per person in Japan than in other countries [7]. Severe MDD is recognized as a high-risk factor for suicide, and its severity is associated with unemployment and absenteeism. Consequently, MDD in workers has become a social problem [8–11]. Therefore, its early detection and appropriate care, including patient education, smooth communication, and listening to patient complaints, are important for reducing unemployment, absenteeism, and presenteeism [12–14].

MDD is often diagnosed based on the criteria listed in the Diagnostic and Statistical Manual, Fifth Edition (DSM-5). According to these criteria, MDD severity is evaluated based on the number of depressive symptoms, which are summed up to determine the presence or absence of a depressive episode [15]. Consequently, the DSM-5 considers MDD to be a unidimensional construct. However, Petersen et al. have reported that a two-factor model fits it better than a one-factor unidimensional model [16]. They found that both somatic and nonsomatic factors are involved in major depressive symptoms. Additionally, Tolentino et al. reported that non-somatic symptoms were associated with severe MDD [17]. The severity of MDD predicts its outcome and may be used to determine the appropriate treatment [18]. However, working patients with MDD are often unaware of their illness and its severity and have difficulty seeking help during the early stages. There is a lack of systematic research on the relationship between early depressive symptoms and disease severity in working patients with MDD. Studies on this topic will enable clinicians to perform early detection and treatment of MDD. We hypothesized that initial self-perceived non-somatic symptoms of MDD are related to MDD severity and aimed to clarify this association in this study.

## Methods

### Participants

Between October 2011 and January 2020, 619 outpatients with MDD were examined during their first visit to the Neuropsychiatry Department of Osaka City University Hospital. The medical examinations were conducted at specialized outpatient departments headed by Deguchi and Iwasaki. All the participants were over the age of 20. Individuals who were pregnant; unemployed; diagnosed with schizophrenia, anxiety disorders other than MDD, substance use disorder, intellectual disability, or major medical disorders (such as neurological disorders); and those with a history of a severe suicide attempt were excluded. Finally, 252 participants were assessed for the presence of MDD.

Participants were diagnosed with MDD by psychiatrists with over five years of training using the Mini International Neuropsychiatric Interview, which aids in diagnosing mental illness; the International Classification of Diseases-10; and the Diagnostic and Statistical Manual, Fourth Edition, Text Revision criteria.

The Mini International Neuropsychiatric Interview is a brief, structured interview developed in partnership with general clinicians and psychiatrists in Europe and the United States. The reliability and validity of the Mini International Neuropsychiatric Interview have been established, and we used it to diagnose depressive episodes. Finally, 118 participants with first-onset MDD were enrolled in the study. Data regarding the participants' demographic and occupational characteristics, including age, gender, marital status, and working hours, were

also collected. All the subjects voluntarily consented to participate in this study and understood that there was no penalty for choosing not to participate. After the study was sufficiently explained, written informed consent was obtained from each of them. The participants' records and information were anonymized and de-identified prior to the analysis. The study was approved by the Human Subjects Review Committee at the Osaka City University (approval number: 4245).

### Severity of depressive symptoms

The severity of depression is commonly assessed using depression rating scales [19–22], of which the most frequently used is the Hamilton Depression Rating Scale (HAM-D17) [19].

We rated each participant's MDD severity at the initial visit using the HAM-D17.

The HAM-D17 consists of 17 items. The scores of each item range from 0 to 2 or 0 to 4 according to the severity of somatic and non-somatic symptoms. Individual scores are summed; the total scores range from 0 to 52 and can be interpreted as follows: > 23: very severe, 19–22: severe, 14–18: moderate, and 8–13: mild [23]. Based on these cut-off scores, the participants were classified into MILD (mild to moderate MDD) and SEVERE (severe to very severe MDD) groups.

### Initial self-perceived depressive symptoms

During the first visit, the physician presented the participants the nine diagnostic criteria for MDD, as listed in the DSM-IV, and asked them to recall the depressive symptoms that led them visit the physician. Then, he/she asked them to select the earliest depressive symptom among the following: (1) depressed mood, (2) loss of interest or pleasure, (3) appetite or weight changes, (4) sleep difficulties (insomnia or hypersomnia), (5) psychomotor agitation or retardation, (6) fatigue or loss of energy nearly every day, (7) feelings of worthlessness or excessive guilt, (8) diminished ability to think or concentrate, and (9) suicidality. These symptoms were classified as non-somatic or somatic based on a two-factor model described in a previous study [16]. "Depressed mood," "loss of interest or pleasure," "feelings of worthlessness or excessive guilt," and "suicidality" were considered non-somatic symptoms, while the others were considered somatic symptoms.

### Statistical analysis

Age and working hours were analyzed as continuous variables, and gender (man, woman), marital status (married, single), and severity (MILD, SEVERE) were analyzed as categorical variables. Continuous variables were compared using Student's t-test, and categorical variables were compared using the $\chi^2$ test. We used univariate analyses and multiple logistic regression analyses to estimate the odds ratios with 95% confidence intervals for the associations between age, gender, marital status, working hours, and initial self-perceived depressive symptoms and MDD severity adjusted for age, gender, marital status, and working hours. All statistical analyses were performed using SPSS Statistics version 26 (IBM Corp, Armonk, NY), and the significance level was set at 5%.

## Results

A total of 118 participants were enrolled in this study; 84 (71.2%) were men and 34 (28.8%) were women (Table 1). The mean ± standard deviation of age, number of working hours, and HAM-D score of the participants were 39.7 ± 9.9 years, 10.4 ± 2.4 hours, and 23.1 ± 8.6, respectively. The SEVERE group consisted of 76 patients, and 77.6% of them were men.

**Table 1. Demographic characteristics of the participants (N = 118).**

| Characteristics | Total | MILD group | SEVERE group | p-value |
|---|---|---|---|---|
| **Gender** | | | | 0.10 |
| Men | 84 (71.2) | 25 (59.5) | 59 (77.6) | |
| Women | 34 (28.8) | 17 (40.5) | 17 (22.4) | |
| **Age** | 39.7±9.9 | 41.0±11.1 | 39.2±9.2 | 0.34 |
| **HAM-D score** | 23.1±8.6 | 14.0±3.3 | 28.2±6.0 | < 0.01 |
| **Working hours** | 10.4±2.4 | 10.1±2.1 | 10.5±2.6 | 0.16 |
| **Marital status** | | | | 0.89 |
| Married | 70 (59.3) | 25 (59.5) | 45 (59.2) | |
| Single | 48 (40.7) | 17 (40.5) | 31 (40.8) | |
| **Initial self-perceived depressive symptoms** | | | | < 0.01 |
| Somatic | 59 (50.0) | 29 (69.0) | 30 (39.4) | |
| Non-somatic | 59 (50.0) | 13 (31.0) | 46 (60.6) | |

The values are shown as either number (percentage) or mean± standard deviation.

HAM-D: Hamilton Depression Rating Scale.

MILD: Mild to moderate major depressive disorder.

SEVERE: Severe to very severe major depressive disorder.

Table 2 shows the participants' initial self-perceived symptoms grouped by MDD severity. In both groups, the most common initial self-perceived symptom was "depressed mood," followed by "fatigue or loss of energy nearly every day." In the SEVERE group, 55.3% of the participants initially perceived a depressed mood, as opposed to 28.6% in the MILD group. Sleep difficulties were initially perceived by 19.0% of the participants in the MILD group and 5.3% of the participants in the SEVERE group. Somatic symptoms were initially perceived by 61.9% and 38.2% of the participants in the MILD and SEVERE groups, respectively.

The results of the logistic regression analyses of the associations between socio-demographic variables, initial self-perceived depressive symptoms, and severity of MDD are shown in Table 3. The univariate analysis showed no association between age, gender, marital status, or working hours and severity of MDD. However, non-somatic symptoms were associated

**Table 2. Initial self-perceived depressive symptoms based on MDD severity (N = 118).**

| Self-perceived symptom | MILD group | SEVERE group |
|---|---|---|
| **Depressed mood** | 12 (28.6) | 42 (55.3) |
| **Loss of interest or pleasure** | 0 (0.0) | 2 (2.6) |
| **Appetite or weight changes** | 0 (0.0) | 1 (1.3) |
| **Sleep difficulties** | 8 (19.0) | 4 (5.3) |
| **Psychomotor agitation or retardation** | 6 (14.3) | 6 (7.9) |
| **Fatigue or loss of energy nearly every day** | 11 (26.2) | 17 (22.4) |
| **Feelings of worthlessness or excessive guilt** | 1 (2.4) | 1 (1.3) |
| **Diminished ability to think or concentrate** | 4 (9.5) | 2 (2.6) |
| **Suicidality** | 0 (0.0) | 1 (1.3) |

The values are shown as number (percentage).

MDD: Major depressive disorder.

MILD: Mild to moderate major depressive disorder.

SEVERE: Severe to very severe major depressive disorder.

**Table 3. Logistic regression analyses of MDD severity, depressive symptoms, and covariates.**

|  | Univariate model | | | Adjusted model[a] | | |
|---|---|---|---|---|---|---|
|  | OR | 95% CI | *p*-value | OR | 95% CI | *p*-value |
| **Age (years)** | 0.99 | 0.95–1.02 | 0.33 | 0.99 | 0.95–1.04 | 0.86 |
| **Gender** |  |  |  |  |  |  |
| Men | 1.00 |  |  | 1.00 |  |  |
| Women | 2.36 | 0.93–5.35 | 0.06 | 2.24 | 0.90–5.58 | 0.08 |
| **Marital status** |  |  |  |  |  |  |
| Married | 1.00 |  |  | 1.00 |  |  |
| Single | 0.99 | 0.46–2.12 | 0.89 | 0.92 | 0.36–2.38 | 0.87 |
| **Working hours** | 1.07 | 0.91–1.25 | 0.43 | 1.04 | 0.87–1.25 | 0.66 |
| **Initial self-perceived depressive symptoms (%)** |  |  |  |  |  |  |
| Somatic symptoms | 1.00 |  |  | 1.00 |  |  |
| Non-somatic symptoms | 3.32 | 1.46–7.58 | < 0.01 | 3.35 | 1.47–7.60 | < 0.01 |

MDD: Major depressive disorder.

95% CI: 95% confidence interval, OR: Odds ratio.

a: Adjusted for age, gender, marital status, working hours, and initial self-perceived depressive symptoms.

with increased odds of severe MDD (odds ratio = 3.32, 95% confidence interval 1.46–7.58), and this association persisted in the adjusted model (odds ratio = 3.35, 95% confidence interval 1.47–7.60).

## Discussion

Our results show that the initial self-perceived, non-somatic depressive symptoms of patients with first-onset MDD are significantly associated with its severity even after controlling for socio-demographic factors. This is the first study to examine the association between initial depressive symptoms and MDD severity at its first onset in working patients.

### Depressive symptoms and MDD severity

Our study demonstrated that initial self-perceived, non-somatic depressive symptoms are associated with MDD severity. A study of 1210 outpatients with MDD at a single institution in the United States found differences among symptoms in their association with MDD severity. Additionally, they found that suicidal ideation, depressed mood, and anhedonia had the highest associations with MDD severity [24]. Another study conducted on 189 outpatients with MDD at two hospitals in Brazil suggested that non-somatic symptoms were associated with severe MDD [17].

We focused on the initial self-perceived depressive symptoms experienced by patients before they were diagnosed with MDD based on the DSM-5 criteria. At this point, they were likely to be in a subthreshold depressive state. Previous studies [17,24] in which MDD was diagnosed based on the DSM-5 criteria have examined depressive symptoms at the time of consultation. The crucial difference between our study and the previous ones is the timing of assessment of perceived depressive symptoms. To our knowledge, our study is the first to highlight that the earliest subjective symptoms felt around the MDD onset may affect its severity. Non-somatic symptoms may be associated with MDD severity, irrespective of whether their symptoms exist at disease onset or at the time of initial diagnosis. Especially, regardless of when non-somatic symptoms exist, their presence may be related to MDD severity. In contrast to our findings, Faravele et al. found that impaired concentration was one of the strongest

predictors of MDD severity [25]. This discrepancy is possibly due to the difference in the classification of depressive symptoms. While we categorized impaired concentration as a somatic symptom, other studies categorized it as a non-somatic symptom [26]. Whether impaired concentration should be considered a somatic or non-somatic symptom needs further investigation.

Depressed mood and loss of interest have been associated with decreased health-related quality of life in patients with MDD [27]. A previous study demonstrated that depressed mood negatively influences psychological states and objective task performance; participants with depressed mood exerted greater mental effort to perform a task than those not in a depressed mood [28]. In addition, depressed mood has been associated with poor sleep quality and suicidal ideation [29,30]. The DSM-5 classifies MDD severity based on the number and intensity of depressive symptoms, degree of functional disability, and distress level. Non-somatic symptoms are associated with other depressive symptoms and poor social functioning, further increasing depression. Consequently, non-somatic symptoms can lead to severe MDD. Determining the severity of MDD is important for predicting its outcome and selecting the appropriate treatment strategy [18]. Therefore, it is important that patients with MDD have a work environment where they can consult with their supervisors and colleagues regarding their depressive symptoms. In addition, supervisors and colleagues might notice depressive symptoms even before the workers become aware of them. Since non-somatic symptoms may be expressed through the individual's facial expressions and tone of voice, workplace support may lead to early detection and treatment of MDD.

## Limitations

This study has some limitations. First, this study was conducted at a single institution (a university hospital). Our findings, therefore, may not be representative of the general population and may have limited generalizability. Second, we evaluated the participants' initial self-perceived symptoms, which required remembering past symptoms; hence, recall bias cannot be ruled out. Third, the study only included Japanese participants, and several cultural factors may have influenced their symptoms. Fourth, while MDD generally has a female predominance, the proportion of male patients in this study was high; therefore, our findings may not accurately reflect the actual situation. Finally, we classified the symptoms as non-somatic or somatic based on the definitions provided in previous studies. However, the classification of symptoms has not been strictly defined and varies between studies. We believe that further research is needed to establish a consistent categorization of depressive symptoms.

## Conclusions

Initial self-perceived non-somatic depressive symptoms are significantly associated with disease severity in working patients with first-onset MDD. Patients with non-somatic symptoms may require more careful treatment than patients with somatic symptoms.

## Acknowledgments

We gratefully acknowledge the work of past and present members of our laboratory.
We would like to thank Editage (www.editage.com) for English language editing.

## Author Contributions

**Conceptualization:** Tomoyuki Hirota, Yasuhiko Deguchi, Shinichi Iwasaki.

**Data curation:** Tomoyuki Hirota, Yasuhiko Deguchi, Shinichi Iwasaki, Aya Sakaguchi, Akihiro Niki, Yoshiki Shirahama, Yoko Nakamichi, Koki Inoue.

**Formal analysis:** Tomoyuki Hirota, Yasuhiko Deguchi, Shinichi Iwasaki.

**Investigation:** Tomoyuki Hirota, Yasuhiko Deguchi, Shinichi Iwasaki.

**Methodology:** Tomoyuki Hirota, Yasuhiko Deguchi, Shinichi Iwasaki.

**Project administration:** Koki Inoue.

**Software:** Tomoyuki Hirota, Yasuhiko Deguchi.

**Supervision:** Shinichi Iwasaki.

**Writing – original draft:** Tomoyuki Hirota, Yasuhiko Deguchi.

**Writing – review & editing:** Yasuhiko Deguchi, Shinichi Iwasaki, Aya Sakaguchi, Akihiro Niki, Yoshiki Shirahama, Yoko Nakamichi, Koki Inoue.

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
