## [Decision Letter · Decision Letter 0]

24 May 2021

PONE-D-21-13717

Relationship between initial self-perceived depressive symptoms and disease severity in working patients with first-onset major depressive disorder

PLOS ONE

Dear Dr. Deguchi,

Thank you for submitting your manuscript to PLOS ONE. After careful consideration, we feel that it has merit but does not fully meet PLOS ONE’s publication criteria as it currently stands. Therefore, we invite you to submit a revised version of the manuscript that addresses the points raised during the review process.

Two reviewers addressed several minor concerns about your manuscript. Please revise your manuscript carefully.

We look forward to receiving your revised manuscript.

Kind regards,

Kenji Hashimoto, PhD

Academic Editor

PLOS ONE

Journal Requirements:

Reviewers' comments:

Reviewer's Responses to Questions

**Comments to the Author**

1. Is the manuscript technically sound, and do the data support the conclusions?

Reviewer #1: Yes

Reviewer #2: Yes

2. Has the statistical analysis been performed appropriately and rigorously? 

Reviewer #1: Yes

Reviewer #2: Yes

3. Have the authors made all data underlying the findings in their manuscript fully available?

Reviewer #1: Yes

Reviewer #2: Yes

4. Is the manuscript presented in an intelligible fashion and written in standard English?

Reviewer #1: Yes

Reviewer #2: Yes

5. Review Comments to the Author

Reviewer #1: In this study the authors examined which initially perceived depressive symptoms predicted later disease severity in working patients with first-onset MDD. The results showed that non-somatic depressive symptoms were associated with severe MDD. The results of this paper are clearly presented and seem to be clinically suggestive. Overall, the manuscript is well-written, and the English is good enough to be read. The paper is of interest for the journal. However, I have some shortcomings to mention as follows;

1. The authors asked patients to report the first depressive symptom they subjectively felt, but it would be an overstatement to label it as a depressive symptom before the onset of MDD symptoms. (Page 14, Line 204-206)

2. The authors state that non-somatic symptoms may be related to the severity of depression, whether the patient is aware of it or not. In the following paragraph, they do not discuss whether people are aware of their depressive symptoms or not. (Page 14-15, Line 207-214)

3. The authors state that the presence or absence of non-somatic symptoms may affect the treatment strategies, but this seems rather abrupt. The rationale for this should be stated. (Page 15, Line 226-227)

Reviewer #2: To the authors

This study aimed to clarify the association between initial depressive symptoms and MDD severity in working patients. The aim of the study is clinically significant and unique.

There are some points to be clarified.

This study investigated the relationship between early depressive symptoms (or initial self-perceived non-somatic symptoms) and disease severity in working patients with MDD. However, it is not clear when early depressive symptoms occurred. The severity of depressive symptoms was evaluated at the initial visit by HAM-D17. When did initial self-perceived depressive symptoms arise? Did patients recall the first one symptom or the first two or more symptoms in their index depressive episodes? The definition of initial self-perceived depressive symptoms (When, How, What, etc.) is not clearly described and should be described in the Abstract, Introduction, and Methods.

In the Methods section, "fatigue" belongs to non-somatic symptoms by citing Petersen et al. However, Petersen et al. regarded "fatigue" as somatic symptoms. I also feel "fatigue" as somatic symptoms. Did the authors make a mistake? If so, the results should be re-analyzed.

Abstract: In "between initial depressive symptoms and MDD severity in working patients", "and" was lost.

6. PLOS authors have the option to publish the peer review history of their article (what does this mean?). If published, this will include your full peer review and any attached files.

Reviewer #1: **Yes: **Hiroyuki Toda

Reviewer #2: No

---

## [Author Response · Author response to Decision Letter 0]

25 Jun 2021

RESPONSES TO REVIEWER #1: 

The authors would like to thank the reviewer for his constructive critique and for providing these insightful comments, which have helped us significantly improve the quality of our work. We have made every effort to address the issues raised and to respond to all comments. The revisions are indicated in red font in the revised manuscript. Please, find next a detailed, point-by-point response to the reviewer's comments.

Comment 1: The authors asked patients to report the first depressive symptom they subjectively felt, but it would be an overstatement to label it as a depressive symptom before the onset of MDD symptoms. (Page 14, Line 204-206)

Response 1: We agree with the reviewer’s comment. Therefore, we have toned down our statement and we have revised this sentence as follows (Lines 206–208): 

From 

“For the first time, our study highlights that early depressive symptoms that occur before the onset of MDD may affect its severity.” 

to 

“To our knowledge, our study is the first to highlight that the earliest subjective symptoms felt around the MDD onset may affect its severity.”

Comment 2: The authors state that non-somatic symptoms may be related to the severity of depression, whether the patient is aware of it or not. In the following paragraph, they do not discuss whether people are aware of their depressive symptoms or not. (Page 14-15, Line 207-214).

Response 2: We agree with the reviewer that this point requires clarification. Following the reviewer’s suggestion, we have added the following part to the Discussion section of the revised manuscript (Lines 208–211)

From

“Non-somatic symptoms may be associated with MDD severity, irrespective of whether the patient is aware of them at disease onset or at the time of initial diagnosis.”

to

“Non-somatic symptoms may be associated with MDD severity, irrespective of whether their symptoms exist at disease onset or at the time of initial diagnosis. Especially, regardless of when non-somatic symptoms exist, their presence may be related to MDD severity.”

Comment 3: The authors state that the presence or absence of non-somatic symptoms may affect the treatment strategies, but this seems rather abrupt. The rationale for this should be stated. (Page 15, Line 226-227)

Response 3: We agree with the reviewer that this point requires clarification. Please note that we have toned down our statement and removed the following part from the Discussion section (Lines 227–228)

This specific part has been revised as follows:

From

“Determining the severity of MDD is important for predicting its outcome and selecting the appropriate treatment strategy. Furthermore, treatment strategies may differ between patients with non-somatic and somatic symptoms.”

to

“Determining the severity of MDD is important for predicting its outcome and selecting the appropriate treatment strategy [18].”

 

RESPONSE TO REVIEWER #2: 

The authors would like to thank the reviewer for his/her constructive critique and for providing these insightful comments, which have helped us significantly improve the quality of our work. We have made every effort to address the issues raised and to respond to all comments. The revisions are indicated in red font in the revised manuscript. Please, find next a detailed, point-by-point response to the reviewer's comments.

Comment 1: This study investigated the relationship between early depressive symptoms (or initial self-perceived non-somatic symptoms) and disease severity in working patients with MDD. However, it is not clear when early depressive symptoms occurred. The severity of depressive symptoms was evaluated at the initial visit by HAM-D17. When did initial self-perceived depressive symptoms arise? Did patients recall the first one symptom or the first two or more symptoms in their index depressive episodes? The definition of initial self-perceived depressive symptoms (When, How, What, etc.) is not clearly described and should be described in the Abstract, Introduction, and Methods.

Response 1: We agree that this point requires clarification, and have added the following text to the Methods. (page 7, lines 114-117)

From

“The participants were asked to select their initial self-perceived depressive symptoms 

from a list based on the following nine DSM-5 diagnostic criteria for MDD. “

to

“During the first visit, the physician presented the participants the nine diagnostic criteria for MDD, as listed in the DSM-Ⅳ, and asked them to recall the depressive symptoms that led them visit the physician. Then, he/she asked them to select the earliest depressive symptom among the following. “

It is also noted in the Limitations that there may be a recall bias since patients were asked to remember past symptoms. (page 16, lines 238-240)

Moreover, we apologize for the mistake in writing "DSM-5" when we should have written "DSM-Ⅳ". This was corrected throughout the manuscript.

Comment 2: In the Methods section, "fatigue" belongs to non-somatic symptoms by citing Petersen et al. However, Petersen et al. regarded "fatigue" as somatic symptoms. I also feel "fatigue" as somatic symptoms. Did the authors make a mistake? If so, the results should be re-analyzed.

Response 2: We apologize for a mistake in writing "fatigue or loss of energy nearly every day" when we should have written "feelings of worthlessness or excessive guilt."

This error has been corrected in accordance with the reviewer's comment.

Nevertheless, the correct data was used in the statistical analysis.

(page 7, lines 122-124)

From

“Depressed mood,” “loss of interest or pleasure,” “fatigue or loss of energy nearly every day,” and “suicidality” were considered non-somatic symptoms, while the others were considered somatic symptoms.”

to

“Depressed mood,” “loss of interest or pleasure,” “feelings of worthlessness or excessive guilt,” and “suicidality” were considered non-somatic symptoms, while the others were considered somatic symptoms.”

Comment 3: Abstract: In "between initial depressive symptoms and MDD severity in working patients", "and" was lost.

Response 3: This error has been corrected in accordance with the reviewer's comment.

---

## [Decision Letter · Decision Letter 1]

12 Jul 2021

Relationship between initial self-perceived depressive symptoms and disease severity in working patients with first-onset major depressive disorder

PONE-D-21-13717R1

Dear Dr. Deguchi,

We’re pleased to inform you that your manuscript has been judged scientifically suitable for publication and will be formally accepted for publication once it meets all outstanding technical requirements.

Kind regards,

Kenji Hashimoto, PhD

Section Editor

PLOS ONE

Additional Editor Comments (optional):

Reviewers' comments:

Reviewer's Responses to Questions

**Comments to the Author**

1. If the authors have adequately addressed your comments raised in a previous round of review and you feel that this manuscript is now acceptable for publication, you may indicate that here to bypass the “Comments to the Author” section, enter your conflict of interest statement in the “Confidential to Editor” section, and submit your "Accept" recommendation.

Reviewer #1: All comments have been addressed

Reviewer #2: All comments have been addressed

2. Is the manuscript technically sound, and do the data support the conclusions?

Reviewer #1: Yes

Reviewer #2: Yes

3. Has the statistical analysis been performed appropriately and rigorously? 

Reviewer #1: Yes

Reviewer #2: Yes

4. Have the authors made all data underlying the findings in their manuscript fully available?

Reviewer #1: Yes

Reviewer #2: Yes

5. Is the manuscript presented in an intelligible fashion and written in standard English?

Reviewer #1: Yes

Reviewer #2: Yes

6. Review Comments to the Author

Reviewer #1: In this study the authors examined which initially perceived depressive symptoms predicted later disease severity in working patients with first-onset MDD. The results showed that non-somatic depressive symptoms were associated with severe MDD. In revised version, the authors have fulfilled all my requirements.

Reviewer #2: To the authors

The authors responded to the reviewers' comments adequately. Their manuscript is written well.

7. PLOS authors have the option to publish the peer review history of their article (what does this mean?). If published, this will include your full peer review and any attached files.

Reviewer #1: **Yes: **Hiroyuki Toda

Reviewer #2: **Yes: **Takeshi Inoue

---

## [Editor Report · Acceptance letter]

14 Jul 2021

PONE-D-21-13717R1 

Relationship between initial self-perceived depressive symptoms and disease severity in working patients with first-onset major depressive disorder 

Dear Dr. Deguchi:

I'm pleased to inform you that your manuscript has been deemed suitable for publication in PLOS ONE. Congratulations! Your manuscript is now with our production department. 

Kind regards, 

on behalf of

Prof. Kenji Hashimoto 

Section Editor

PLOS ONE